# Effects of Different Parts on the Chemical Composition, Silage Fermentation Profile, In Vitro and In Situ Digestibility of Paper Mulberry

**DOI:** 10.3390/ani11020413

**Published:** 2021-02-05

**Authors:** Yangyi Hao, Shuai Huang, Gaokun Liu, Jun Zhang, Gang Liu, Zhijun Cao, Yajing Wang, Wei Wang, Shengli Li

**Affiliations:** State Key Laboratory of Animal Nutrition, Beijing Engineering Technology Research Center of Raw Milk Quality and Safety Control, College of Animal Science and Technology, China Agricultural University, Beijing 100193, China; haoyangyi0928@163.com (Y.H.); huangshuai510@126.com (S.H.); liugk339@163.com (G.L.); june_zh16@cau.edu.cn (J.Z.); liugang_0402@126.com (G.L.); caozhijun@cau.edu.cn (Z.C.); yajingwang@cau.edu.cn (Y.W.); wei.wang@cau.edu.cn (W.W.)

**Keywords:** paper mulberry, nutritional value, silage, digestibility

## Abstract

**Simple Summary:**

Paper mulberry (*Broussonetia papyrifera*, PM) is a potential roughage source widely distributed in Asia, but the chemical composition, silage fermentation, and digestibility are not fully understood. Here, we compared the chemical composition, silage fermentation, and digestibility of leaf, stem, and whole plant of PM to evaluate its feeding value. The result showed that the leaf had lower fiber content and higher protein content than the stem and whole plant. Meanwhile, the stem silage had the lowest pH value and lactate content, while those in the leaf were the highest. The in vitro and in situ digestibility showed the leaf was more digestible. Our study gives the reference of different parts of PM to be used as a feedstuff.

**Abstract:**

Paper mulberry (*Broussonetia papyrifera*, PM) is high protein but unutilized as a feed source. The study explores the different parts (leaf, stem, and whole plant) of PM chemical composition, silage fermentation, and in vitro and in situ digestibility, aiming to give some guidelines to PM usage as feed. The result showed that the leaf had a higher fresh weight than the stem (*p* < 0.05). The dry matter contents of the three groups had no differences. The highest crude protein, ether extract, water-soluble carbohydrate, ash, calcium, phosphorus, amino acid contents, and butter capacity were observed in the leaf (*p* < 0.05). The stem had the highest (*p* < 0.05) neutral detergent fiber, acid detergent fiber, and lignin contents. After ensiling, the stem silage had the lowest pH value, ammonia nitrate (NH_3_-N), lactate, acetate, and propionate (*p* < 0.05). The leaf silage had the highest pH value (*p* < 0.05). The lactate, acetate, and propionate in the leaf and whole plant silage had no difference. The butyrate was not detected in all silage. The in vitro and in situ digestibility experiments showed the leaf had the highest digestibility (*p* < 0.05), which could produce more volatile fatty acids and have a higher effective digestibility. These results allow a greater understanding of PM to be used as a feedstuff.

## 1. Introduction

The shortage of feed resources and high feeding costs have become important factors restricting the development of the dairy farming industry. Exploitation of new feed resources may be an effective solution for this dilemma. Paper mulberry (*Broussonetia papyrifera*, PM) is a dioecious plant native to mainland Southeast Asia and East Asia [1]. Owing to its strong rooting ability and rapid growth, PM is considered a plant capable of providing environmental protection in many areas [2]. As one of China’s top ten targeted poverty alleviation projects, China has planted over 300,000 hectares of PM, which serves as a potential animal feed source with economic and functional merits. The bark of PM is a good source of fiber for the production of paper. The mulberry fruit contains many bioactive chemical compounds, such as alkaloids, flavonoids, anthocyanins, and some polyphenols, strengthening animals’ immunity and antioxidant capacity [3]. It was reported that the extracts of mulberry could ameliorate inflammation in rats [4], and the anthocyanins from mulberry fruits can scavenge free radicals and inhibit low-density lipid oxidation [5]. Owing to its high protein concentration in the leaf, PM can also be used as a feedstuff for animals [6]. A previous study found that dietary supplementation with mulberry leaf powder can improve digestion in ruminants [7]. Diets with 10% or 15% whole plant PM silage, which is substituted for alfalfa hay, could reportedly reduce the milk somatic cell count and enhance the antioxidant capacity and immune function of lactating dairy cows, but also decreased the feed intake [8].

A good feedstuff should be conveniently conserved and utilized. Currently, silage is the best conserving technique because it is affordable and easy to make, which also has the advantage of minimum loss of nutrients. However, to our knowledge, no information is available on the PM chemical composition and silage fermentation profile. The reports of PM progressing, conserving, and using are also lacking. All these limit PM being used as a feedstuff. To understand PM fully, we compared the leaf, stem, and whole plant nutrition of PM. Therefore, the study’s objective was to explore the effects of different parts on the chemical composition, silage fermentation profile, and in vitro and in situ digestibility of PM.

## 2. Materials and Methods

### 2.1. Harvesting and Different Fractions of the Plants

The PM was planted in Henan province, which is in central China. The area’s longitude and latitude are 114.82° and 34.82°, respectively, which belong to a monsoon climate of medium latitudes with an average annual temperature of 15.7 °C and an average yearly rainfall of 485 mm in the last three years. The field was sandy loam and weeds were previously growing there. A total of 70 kg/ha N fertilizer was applied annually in the field. The PM was seeded using transplant seedlings, which can be harvested continuously for about fifteen years and can harvest three to five times a year. The PM was harvested from three fields at a 1.2 m growth height and a cutting height of 0.2 m in May 2018. First, twenty plants were collected in each field and weighted. We separated twelve plants into leaf and stem and weighted, respectively. The remaining plants served as the whole plant. Each field plant is one replicate, and we have three replicates in all.

### 2.2. Nutritional Composition of Fresh Samples

The PM was treated as described by Gallo et al. [9]. Briefly, the samples were immediately placed into a forced-draft oven (DGG-9240B; Shanghai-ShenXin Inc., Shanghai, China) at 65 °C for 48 h to determine dry matter (DM). Samples were then milled through a 1 mm screen using a rotating-feedstuff mill (KRT-34; KunJie, Beijing, China) and stored at ambient temperature before subsequent analysis. The neutral detergent fiber (NDF) and acid detergent fiber (ADF) were determined using an ANKOM fiber analyzer (A2000i; American ANKOM, NY, USA), as described by Van Soest et al. [10]. The lignin was measured with reference to Theander et al. [11]. Nitrogen was measured according to method 984.13 of the Association of Official Analytical Chemists (AOAC) [12]. Crude protein (CP) was calculated by multiplying 6.25 by the nitrogen content. Water-soluble carbohydrate (WSC) was measured using the anthrone method described by Murphy et al. [13]. Ash (method 924.05) and ether extract (EE, method 920.39) were measured according to the methods of the AOAC [12]. The buffering capacity of PM was measured according to the sodium hydroxide titration method [14]. The amino acid composition of PM was measured using high-performance liquid chromatography (HPLC) (Agilent 1200, Agilent Technologies, Everett, WA, USA) as previously described by Li et al. [15].

### 2.3. PM Silage

#### 2.3.1. Silage Making

After harvesting, separation, and weighing, the leaf, stem, and whole plant PM from three different fields were cut into 1–2 cm using a manual forage chopper (93ZT-300; Xingrong Co., Ltd. Guangzhou, China), respectively. The cut pieces were then packed into plastic polyethylene bottles (0.5 L capacity). Air was removed from the bottles, which were then sealed using a vacuum sealer. All silage samples had a density higher than 800 kg/m^3^ and were stored at ambient temperature for 45 days.

#### 2.3.2. Silage Chemical Composition Determination

In each bottle, 20 g of silage was blended with 180 mL deionized water and stored at 4 °C for 24 h [16]. The mixture was then filtered through four layers of cheesecloth to determine the fermentation parameters. The silage’s pH was measured immediately using a pH electrode after the filtering (Model pH B-4; Shanghai Chemical, Shanghai, China). The NH_3_-N concentration was measured using the phenol-sodium hypochlorite colorimetry method described by Broderick et al. [17]. The organic acid content was measured using HPLC (Agilent 1200, Agilent Technologies, Everett, WA, USA), according to Yuan et al. [18]. The remaining sample was used for nutritional value detection. The pretreatment, DM, NDF, ADF, Ash, CP, and EE detection methods were the same as those described in the fresh PM chemical composition detection.

### 2.4. In Vitro Digestibility

The leaf, stem, and whole plant PM silage were taken for in vitro digestibility experiment. The in vitro gas production was determined using an Automated Trace Gas Recording System (AGRS) for microbial fermentation, as described by Bai et al. [19,20]. Briefly, 500 mg (DM basis) of representative samples (four replicates) of each treatment group were weighed into 120 mL glass bottles, and 50 mL of a freshly prepared buffer solution was added to each bottle [20]. The components of the buffer solution are presented in Appendix A. Rumen fluid was collected from three fistulated Holstein dairy cows before morning feeding, stored in a vacuum flask, and immediately taken to the lab. All cows were in a state of peak lactation, and the formula of the diets is presented in Appendix A. Before usage, rumen fluid was filtered through four-layer cheesecloth and mixed. A volume of 25 mL of filtered rumen fluid was added into the 120 mL glass bottles. The bottles were then purged with anaerobic N_2_ for 5 s, sealed with a butyl rubber stopper, and individually connected with medical plastic infusion pipes to the gas inlets of the AGRS to record cumulative gas production continuously. The measurement of pH and NH_3_-N was the same as described in the silage fermentation test. The in vitro dry matter digestibility (IVDMD) was calculated using differential subtraction according to the dry matter content of substrates before and after in vitro incubation. Volatile fatty acids (VFA) were determined using HPLC (Agilent 1200, Agilent Technologies, Everett, WA, USA) [21].

### 2.5. In Situ Digestibility

The in situ DM, CP, NDF, and ADF degradation rates of the leaf, stem, and whole plant silage were determined by reference to Nocek et al. [22]. The dried samples were milled through a 3-mm sieve, and 5 g of the prepared sample was weighed and placed in nylon bags (45-mm pore size, 8 × 16 cm bag size) in six repetitions. These samples were incubated for 4, 8, 12, 24, 30, 36, 48, and 72 h in the rumen of three fistulated cows (each cow had two repetitions). The cows and diet were the same as those in the in vitro experiment.

After removing the bags at each time point, the bags were washed in running tap water until the outlet water became clear. Bags were then dried to a constant weight at 65 °C for 48 h and weighed. The residues were ground through a 1 mm sieve for further analysis. The DM, CP, NDF, and ADF were measured as described before.

### 2.6. Calculation and Statistical Analysis

#### 2.6.1. Calculation of Gas Production Parameters for In Vitro Fermentation

The cumulative gas production (*GP*_48_) (mL/g) data were recorded using the AGRS system and fitted to the Groot model as Equation (1) [23]:(1)$$GP_{t}=A/[1+{\left(\frac{C}{t}\right)}^{B}]$$

*A* is the asymptotic gas production (mL/g); *B* is a sharpness parameter determining the curve’s shape; *C* is the time (h) at which half of *A* is reached; and t is in vitro incubation time (h).

The time at which maximum rate of substrate degradation is reached (*TRmaxS*, h), the maximum rate of substrate digestion (*RmaxS*, h), the time at which RmaxG is reached (*TRmaxG*, h), and the maximum gas production rate (*RmaxG*, mL/h) were calculated with *A*, *B*, and *C* as Equation (2)–(5) [24]:(2)$$TRmaxS=C\times {\left(B-1\right)}^{\left(1/B\right)}$$
(3)$$RmaxS=\left[B\times TRmaxS^{\left(B-1\right)}\right]/\left(C^{B}+TRmaxs^{B}\right)$$
(4)$$TRmaxG=C\times {\left[\left(B-1\right)/\left(B+1\right)\right]}^{\left(1/B\right)}$$
(5)$$RmaxG=\left[A\times C^{B}\times B\times TRmaxG^{\left(-B-1\right)}\right]/{\left[1+C^{B}\times TRmaxG^{\left(-B\right)}\right]}^{2}$$

#### 2.6.2. Calculation of in Situ Digestibility

The degradation data were fitted to the following exponential equation [25]:𝑦 = 𝑎 + 𝑏 (1 − 𝑒 ^− 𝑐𝑡^)(6)
*y* is the nutrient disappearance rate in the rumen at time t; *a* is the rapidly degradable fraction; *b* is the potentially degradable fraction; and *c* is the constant rate of degradation of *b* (%/h).

The effective degradability (ED) of nutrients was calculated by applying the following equation [25]:(7)$$ED=a+(\frac{bc}{c+k})$$
*a*, *b*, and *c* are the same parameters represented in Equation (6), and *k* is the rumen outflow rate. The *ED* of nutrients was calculated using an outflow rate of 0.07/h, according to the report of Batajoo and Shaver [26].

#### 2.6.3. Statistical Analysis

Analyses were performed using SAS (SAS version 9.4, SAS Institute Inc., Cary, NC, USA). All data were subjected to one-way analysis of variance (ANOVA). The statistical model was as follows:(8)$$Y_{ij}=\;µ+T_{i}+E_{ij}$$

*Y**_ij_* represents the observed dependent variables; μ is the overall mean; *T**_i_* is the effect of treatment; and *E**_ij_* is the residual error. Statistical differences between means were determined using Duncan’s multiple comparison test. Differences were considered significant when the *p*-value was less than 0.05.

## 3. Results

### 3.1. Yield and Chemical Composition

The leaf had a higher fresh weight (FW) yield than the stem (*p* < 0.05) (Table 1), and the leaf:stem ratio was 1.45:1. The leaf had the highest (*p* < 0.05) CP, EE, WSC, ash, calcium, phosphorus, and buffer capacity, and the lowest (*p* < 0.05) NDF, ADF, and lignin. The highest (*p* < 0.05) NDF, ADF, and lignin and the lowest (*p* < 0.05) CP, WSC, ash, calcium concentration, and buffer capacity was observed in the stem. The different parts of PM did not affect (*p* > 0.05) DM.

The stem and leaf had the lowest (*p* < 0.05) and highest (*p* < 0.05) detected amino acid, respectively (Table 2). Notably, the lysine and methionine in the leaf were 13.5 and 3.9 g/kg, respectively, with a ratio approximating 3:1. The top five amino acids in the leaf, stem, and whole plant were aspartic acid, glutamic acid, leucine, proline, and lysine. The bottom five amino acids were cysteine, tryptophan, methionine, histidine, and tyrosine.

### 3.2. Silage Fermentation Profile

The CP and pH were the highest (*p* < 0.05) and the NDF and ADF were the lowest (*p* < 0.05) in the leaf silage (Table 3). The CP, pH, and acetate in the stem silage were significantly lower (*p* < 0.05) than leaf silage and whole plant silage, while NDF and ADF were significantly higher (*p* < 0.05) than the other groups. The pH, NH_3_-N, lactate, acetate, propionate and butyrate in whole plant PM silage were 4.80, 43.94 (g/kg TN), 90.65, 22.71, 23.96, and 0 (g/kg DM). There is no significant difference of NH_3_-N, lactate, and propionate between leaf silage and whole plant silage, but they were significantly higher (*p* < 0.05) than the stem silage. The butyrate was too low to be detected in all silage samples. The stem silage also did not have propionate be detected.

### 3.3. In Vitro Digestibility

The leaf silage had the highest IVDMD, while the stem silage was the lowest (*p* < 0.05) (Table 4). The IVDMD of the whole plant PM silage was 683.51 (g/kg DM). Acetate concentration was significantly lower (*p* < 0.05) in the stem silage than the leaf and whole plant silage. Compared to the stem and whole plant silage, the leaf silage had a higher (*p* < 0.05) propionate content. Total VFA in the leaf silage was significantly higher (*p* < 0.05) than the stem silage. There was no distinction (*p* > 0.05) of pH and NH_3_-N within the three groups.

The GP_48_, the time (h) at which half of asymptotic gas production is reached (parameters *C*), RmaxG, TRmaxS, RmaxS, and TRmaxG had no significant distinction (*p* > 0.05) within the three different silage. The asymptotic gas production (mL/g) (parameter *A*) and sharpness parameter determining the shape of the curve (parameter *B*) in the leaf silage were higher (*p* < 0.05) than other silage.

### 3.4. In Situ Digestibility

The degradation rate was the highest in leaf silage and lowest in stem silage (*p* < 0.05) (Figure 1). The EDs of DM, CP, NDF, and ADF in the leaf silage were the highest, and those of the stem silage were the lowest (*p* < 0.05) (Table 5). The EDs of DM, CP, NDF, and ADF in leaf, stem, and whole plant PM silage were 621.96, 208.92, and 439.59; 612.85, 191.74, and 455.05; 526.38, 244.92, and 412.93; 500.23, 280.17, and 381.29; respectively. The rapid degradation fraction (parameter *a*) of DM and CP in the leaf silage was highest (*p* < 0.05), while that of ADF showed no significant difference (*p* > 0.05) within the different parts. The leaf silage showed the highest (*p* < 0.05) potential degradation fraction (parameter *b*) of NDF and ADF. Potential degradation fractions of DM, CP, and ADF in the stem group were lower (*p* < 0.05) than those in the other two groups. In contrast, the potential degradation fractions of NDF in the stem group showed no significant difference (*p* > 0.05) from the whole plant group. The CP and NDF constant rate of slow degradation fraction (parameter *c*) were the highest (*p* < 0.05) in the leaf silage, while that of DM in the whole plant and leaf silage was higher (*p* < 0.05) than the stem silage.

## 4. Discussion

### 4.1. Yield and Chemical Composition

Previous studies found that forage chemical composition could influence the animal feeding behavior, feed intake, metabolism, and performance [27,28,29]. Fiber is beneficial for rumination and rumen pH; simultaneously, it is negatively correlated with dry matter intake and digestibility [30]. Leaf has less undigestible fiber, which tends to be a widely used roughage source for ruminants [29,30]. Lignin is a complex, phenolic polymer found in plant cell walls essential for mechanical support, water, and mineral transport [31]. Stem, as the support of the whole plant, is lignified during the growth of the plant. Therefore, the higher lignin that was observed in the stem and whole plant in the current study was understandable. With the lower fiber and higher CP, the leaf had a higher nutritional value; it could be recommended to feed the high-lactation dairy cows, while the stem or the whole plant could be provided to the heifers or dry cows. Silage is a great conserving technique for PM. WSC is beneficial for silage making, while the buffer capacity goes against it [32,33]. The higher WSC in leaf was more advantageous for silage making than stem, while the buffer capacity index was contrary in the two parts. Ash is an important factor affecting feed buffer capacity [34]. We speculated that the PM is a broad-leaved plant, and the leaf surface is relatively coarse, which adheres some dust from the air and causes its ash content to be high. The high ash content enhanced the leaf buffer capacity. Based on the high buffer capacity in leaf and low WSC in stem, it is necessary to consider their respective limiting factors when making PM silage.

Balanced amino acid composition of the diet could improve animal production and performance [35]. The first and second limiting amino acids are lysine and methionine for dairy cows, respectively [35,36]. Leaf had the highest amino acid content, and the ratio of lysine:methionine in PM leaf was close to 3:1, which is approximately an ideal protein ratio for the dairy cows [35]. Some studies indicated histidine is another limiting amino acid in lactating dairy cows based on corn silage and alfalfa haylage (the most common diet type in China) [37,38,39]. However, compared to other amino acids, the PM’s histidine content (including leaf, stem, and whole plant) was relatively low, which should be taken into consideration when a PM-based diet is supplemented with concentrates for dairy cows. Among the top five amino acids in the leaf and whole plant, leucine and lysine were the essential amino acids for dairy cows, making up for the deficiency that cows cannot synthesize these amino acids by themselves [40,41]. In short, PM’s amino acid composition does not entirely meet the ideal amino acid requirements of dairy cows. When using PM to make formulas, it needs to combine with the high histidine and methionine feeds to achieve amino acid balance.

### 4.2. Silage Fermentation Profile

High buffer capacity could inhibit the silage pH downregulation [32]. The highest pH value of leaf silage could be attributed to the higher buffer capacity in the leaf. Buffer capacity became the dominant factor in downregulating the PM silage pH values, which offset the leaf’s WSC effects. Si et al. showed that feeding dairy cows PM silage with a pH value of 5.06 did not affect the milk yield performance and body condition score [8], which indicated the PM silage in our study qualified for the feeding of dairy cows. Our results’ NH_3_-N content was similar to legume silage but higher than grass silage [32]. NH_3_-N was the by-product of silage fermentation, and it was produced by protease, accompanied by the decomposition of proteins by microorganisms [42,43]. The leaf and whole plant silage had a higher NH_3_-N content; this may be due to its higher CP content.

Lactate, produced via homofermentative, could downregulate the silage pH values and inhibit harmful bacteria’s growth [32]. Acetate could enhance the stability of silage by inhibiting the second fermentation [32,33]. The higher organic acid content in the leaf and whole plant is a sign of good quality silage, which is beneficial for dairy cows’ production performance [28,33]. However, the discrepancy between the organic acid and pH values in leaf silage may be due to its higher WSC and buffer capacity. There seems to be other unidentified chemical components in PM that conserve it under these conditions, and this should further explore it. Butyrate is ordinarily undetectable in well-fermented silage [44]. The current study testified that PM could be processed into a successful silage with no butyrate being detected. However, to produce better-fermentation (lower pH, NH_3_-N contents, etc.) PM silage, it is better to add some organic acid (like acetate) and molasses to the leaf and stem, respectively [32,33].

### 4.3. Digestibility

The digestibility is affected by feed species, chemical composition, animal DM intake, healthy status, rumen bacteria, etc. [22,45,46]. Our previous studies have shown that adding 4.5–18.0% PM silage (substituting the alfalfa and oat grass) to the diet did not affect the total-tract apparent digestibility of dairy cows [47]. However, in this study, we proved that the different parts of PM had distinct digestibility. The high diet NDF level could decrease the ruminal digestibility [30]. Stem had more fiber, and leaf had higher CP; these different chemical composition influence PM’s digestibility performance. The leaf had the highest IVDMD and in situ digestibility, mainly attributed to its lower fiber content [46,48,49]. The digestibility kinetics is affected by feed particle size, species, and composition [22,50]. Under the same test procedure (particle size, species, etc.), the leaf group had a higher and faster degradation rate proven by the higher theoretical maximum gas production and in site dynamic degradation parameters [46,48]. All these results indicate that the leaf silage was more digestible [48,51,52].

The digested organic matter was fermented to VFA and gas production (GP) or converted to microbial protein [48]. The total GP is an indicator of forage digestibility [53]. In agreement with Getachew et al. [54], the leaf’s lower lignin content contributed to the higher GP. Fermentative gas is mainly produced when feedstuffs are fermented to acetate and butyrate [55]. There is a positive correlation between VFA production and GP [56]. The IVDMD and VFA results were in correspondence with GP in the current in vitro experiment. The higher IVDMD contributed to the higher total VFA content of the leaf, which could provide a more digestible substance to fermentation. The acetate and propionate acid are the primary energy source of ruminants; they are the precursor of milk fat and milk lactose, respectively [27]. Thus the in vitro experiment indicated leaf might be more beneficial for animal production.

## 5. Conclusions

As a new roughage feedstuff, the different parts of paper mulberry had distinct chemical compositions. The leaf was more nutritious and digestible. Considering the low water-soluble carbohydrate and high buffer capacity, adding some additives to enhance the paper mulberry silage fermentation quality is necessary. Mechanically separating the leaf from the stem before mixing with the diet is recommended to improve paper mulberry’s utilization efficiency. Further study is needed to elucidate the effects of different paper mulberry parts on animals’ production performance.

## Figures and Tables

**Figure 1 animals-11-00413-f001:**
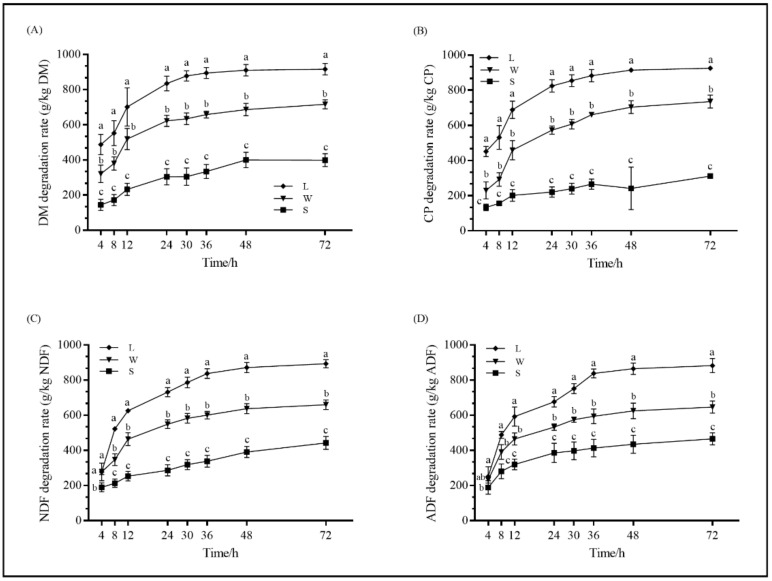
The real-time degradation rates of different parts of PM silage in rumen (**A**), DM, (**B**) CP, (**C**) NDF, and (**D**) ADF. Means within the same time point with different letters are significantly different (*p* < 0.05). L: leaf; S: stem; W: whole plant; DM: dry matter; CP: crude protein; NDF: neutral detergent fiber; ADF: acid detergent fiber.

**Table 1 animals-11-00413-t001:** Fresh weight yield and chemical composition of different parts of paper mulberry.

Items ^1^	Groups		
	Leaf	Stem	Whole Plant
FW yield (kg/per plant)	1.70 ± 0.05 ^b^	1.17 ± 0.04 ^c^	2.79 ± 0.09 ^a^
DM (g/kg)	261.05 ± 4.00	271.06 ± 8.06	266 ± 6.07
CP (g/kg DM)	241.24 ± 0.98 ^a^	84.93 ± 3.29 ^c^	169.40 ± 4.23 ^b^
NDF (g/kg DM)	447.09 ± 31.61 ^c^	680.63 ± 14.60 ^a^	502.79 ± 11.07 ^b^
ADF (g/kg DM)	239.00 ± 13.86 ^c^	565.91 ± 9.81 ^a^	348.88 ± 5.57 ^b^
Lignin (g/kg DM)	33.45 ± 5.79 ^c^	75.67 ± 2.12 ^a^	59.61 ± 3.11 ^b^
EE (g/kg DM)	29.24 ± 0.04 ^a^	22.82 ± 0.25 ^b^	22.54 ± 2.88 ^b^
WSC (g/kg DM)	45.82 ± 4.62 ^a^	11.57 ± 0.77 ^c^	33.24 ± 1.32 ^b^
Ash (g/kg DM)	116.62 ± 0.73 ^a^	47.31 ± 0.41 ^c^	84.95 ± 5.38 ^b^
Calcium (g/kg DM)	26.47 ± 1.79 ^a^	9.09 ± 1.07 ^c^	15.55 ± 0.92 ^b^
Phosphorus (g/kg DM)	3.01 ± 0.11 ^a^	2.50 ± 0.23 ^b^	2.26 ± 0.28 ^b^
Buffer capacity (mE/Kg · DM)	418.30 ± 21.25 ^a^	187.97 ± 5.51 ^c^	297.94 ± 26.86 ^b^

Values with different lowercase superscript letters (a, b, and c) are significantly different (*p* < 0.05) between different parts. ^1^ FW: fresh weight; DM: dry matter; CP: crude protein; NDF: neutral detergent fiber; ADF: acid detergent fiber; WSC: water-soluble carbohydrate; EE: ether extract.

**Table 2 animals-11-00413-t002:** Amino acid composition of different parts of the paper mulberry.

Items	Groups		
	Leaf	Stem	Whole Plant
Animo acid (g/kg DM)			
Tryptophan	3.74 ± 0.04 ^a^	0.83 ± 0.04 ^c^	2.41 ± 0.02 ^b^
Cysteine	2.21 ± 0.05 ^a^	0.51 ± 0.02 ^c^	1.63 ± 0.07 ^b^
Methionine	3.92 ± 0.15 ^a^	1.04 ± 0.04 ^c^	2.53 ± 0.23 ^b^
Aspartic acid	23.42 ± 0.71 ^a^	9.83 ± 0.20 ^c^	17.81 ± 0.43 ^b^
Threonine	11.12 ± 0.12 ^a^	2.74 ± 0.05^c^	7.23 ± 0.11 ^b^
Serine	9.73 ± 0.23 ^a^	2.84 ± 0.05 ^c^	6.89 ± 0.17 ^b^
Glutamic acid	24.68 ± 0.28 ^a^	6.33 ± 0.10 ^c^	16.53 ± 0.49 ^b^
Proline	15.72 ± 0.37^a^	5.30 ± 0.17 ^c^	10.63 ± 0.62 ^b^
Glycine	11.32 ± 0.18 ^a^	2.58 ± 0.07 ^c^	7.59 ± 0.20 ^b^
Alanine	10.29 ± 0.20 ^a^	2.54 ± 0.04 ^c^	7.04 ± 0.18 ^b^
Valine	13.41 ± 0.38 ^a^	3.46 ± 0.04 ^c^	9.03 ± 0.23 ^b^
Isoleucine	10.40 ± 0.21 ^a^	2.45 ± 0.11 ^c^	7.03 ± 0.18 ^b^
Leucine	18.82 ± 0.22 ^a^	4.21 ± 0.15 ^c^	12.52 ± 0.24 ^b^
Tyrosine	7.48 ± 0.18 ^a^	0.91 ± 0.02 ^c^	4.47 ± 0.17 ^b^
Phenylalanine	11.67 ± 0.32 ^a^	2.58 ± 0.10 ^c^	7.66 ± 0.08 ^b^
Histidine	4.51 ± 0.08 ^a^	1.26 ± 0.03 ^c^	3.17 ± 0.12 ^b^
Lysine	13.53 ± 0.28 ^a^	3.48 ± 0.10 ^c^	9.68 ± 0.24 ^b^
Arginine	12.81 ± 0.22 ^a^	2.43 ± 0.04 ^c^	8.27 ± 0.10 ^b^

Values with different superscript letters (a, b, and c) are significantly different (*p* < 0.05) between the different parts.

**Table 3 animals-11-00413-t003:** Chemical composition and fermentation profile of different parts of paper mulberry silage.

Items ^1^	Groups		
	Leaf	Stem	Whole Plant
Chemical Composition			
DM (g/kg)	261.04 ± 4.00 ^ab^	271.06 ± 8.06 ^a^	256.80 ± 1.70 ^b^
CP (g/kg DM)	252.87 ± 11.57 ^a^	92.28 ± 1.90 ^c^	190.76 ± 6.26 ^b^
NDF (g/kg DM)	304.25 ± 6.06 ^c^	670.19 ± 2.68 ^a^	415.45 ± 12.91 ^b^
ADF (g/kg DM)	158.48 ± 14.29 ^c^	580.47 ± 8.57 ^a^	323.72 ± 23.90 ^b^
Fermentation Profile			
pH	5.09 ± 0.06 ^a^	3.81 ± 0.01 ^c^	4.80 ± 0.20 ^b^
NH_3_-N (g/kg TN)	44.07 ± 7.37 ^a^	20.91 ± 3.23 ^b^	43.94 ± 10.87 ^a^
Lactate (g/kg DM)	99.77 ± 2.29 ^a^	52.07 ± 6.62 ^b^	90.65 ± 13.92 ^a^
Acetate (g/kg DM)	39.63 ± 4.71 ^a^	6.43 ± 1.78 ^c^	22.71 ± 2.52 ^b^
Propionate (g/kg DM)	21.03 ± 2.27 ^a^	ND ^b^	23.96 ± 6.03 ^a^
Butyrate (g/kg DM)	ND	ND	ND

Values with different lowercase superscript letters (a, b, and c) are significantly different (*p* < 0.05) between different parts. ^1^ DM: dry matter; CP: crude protein; NDF: neutral detergent fiber; ADF: acid detergent fiber; TN: total N, NH_3_-N/TN: NH_3_-N to total N ratio; ND: no detection.

**Table 4 animals-11-00413-t004:** The IVDMD, in vitro gas production kinetics, and rumen fluid fermentation profile of different parts of paper mulberry silage.

Items ^1^	Groups		
	Leaf	Stem	Whole Plant
IVDMD (g/kg DM)	909.00 ± 22.73 ^a^	484.31 ± 22.16 ^c^	683.51 ± 8.6 ^b^
pH	6.94 ± 0.02	6.93 ± 0.07	6.97 ± 0.05
NH_3_-N (mg/dL)	12.08 ± 0.45	11.10 ± 0.83	11.94 ± 0.30
Acetate (mmol/L)	53.03 ± 0.59 ^a^	48.25 ± 1.79 ^b^	52.99 ± 1.59 ^a^
Propionate (mmol/L)	24.67 ± 1.04 ^a^	19.09 ± 0.98 ^b^	20.07 ± 0.87 ^b^
Isobutyrate (mmol/L)	0.72 ± 0.17 ^b^	1.57 ± 0.08 ^a^	1.67 ± 0.15 ^a^
Butyrate (mmol/L)	9.76 ± 0.76	8.32 ± 0.39	9.28 ± 0.52
Isovalerate (mmol/L)	2.39 ± 0.28	2.04 ± 0.15	2.44 ± 0.21
Valerate (mmol/L)	2.05 ± 0.07 ^a^	1.65 ± 0.01 ^b^	1.92 ± 0.10 ^ab^
Total VFA (mmol/L)	92.63 ± 2.83 ^a^	80.92 ± 1.36 ^b^	88.37 ± 1.89^ab^
GP_48_ (mL/g)	90.05 ± 7.74	74.24 ± 12.28	82.40 ± 8.31
*A*	95.65 ± 7.72 ^a^	75.92 ± 12.53 ^b^	89.56 ± 8.50 ^b^
*B*	1.44 ± 0.10 ^a^	1.28 ± 0.04 ^b^	1.27 ± 0.05 ^b^
*C*	3.95 ± 0.33	4.00 ± 0.57	4.09 ± 0.60
TRmaxG	1.21 ± 0.31	0.78 ± 0.09	0.78 ± 0.25
RmaxG	15.05 ± 2.10	12.28 ± 1.95	13.38 ± 2.60
TRmaxS	2.07 ± 0.54	1.56 ± 0.02	1.60 ± 0.48
RmaxS	0.20 ± 0.02	0.19 ± 0.03	0.19 ± 0.03

Values with different lowercase superscript letters (a, b, and c) are significantly different (*p* < 0.05) between different parts. ^1^ IVDMD: in vitro dry matter digestibility; VFA: volatile fatty acid; GP_48_: the cumulative gas production at 48 h; *A*: the asymptotic gas production (mL/g); *B*: a sharpness parameter determining the shape of the curve; *C*: the time (h) at which half of A is reached and *t* is in vitro incubation time; TRmaxS: the time at which maximum rate of substrate degradation is reached (h); RmaxS: the maximum rate of substrate digestion (/h); TRmaxG: the time at which RmaxG is reached (h); RmaxG: the maximum gas production rate (mL/h).

**Table 5 animals-11-00413-t005:** Parameters of DM, CP, NDF, and ADF dynamic degradation model of different parts of paper mulberry silage.

Items ^1^	Groups		
	Leaf	Stem	Whole Plant
**DM**			
*a* (g/kg DM)	337.91 ± 65.14 ^a^	104.96 ± 7.63 ^b^	147.74 ± 39.18 ^b^
*b* (g/kg DM)	601.13 ± 43.94 ^a^	345.70 ± 37.62 ^b^	544.90 ± 29.24 ^a^
*c* (g/kg DM)	63.09 ± 11.86 ^a^	30.71 ± 4.63 ^b^	78.72 ± 4.98 ^a^
*a* + *b* (g/kg DM)	939.04 ± 23.73 ^a^	450.66 ± 34.85 ^c^	702.64 ± 15.41 ^b^
ED (g/kg DM)	621.96 ± 19.15 ^a^	208.92 ± 6.12 ^c^	439.59 ± 27.43 ^b^
**CP**			
*a* (g/kg DM)	306.21 ± 41.64 ^a^	155.68 ± 16.15 ^b^	174.71 ± 5.73 ^b^
*b* (g/kg DM)	626.80 ± 32.98 ^a^	293.78 ± 99.84 ^b^	629.84 ± 29.96 ^a^
*c* (g/kg DM)	89.14 ± 1.97 ^a^	50.47 ± 7.34 ^b^	59.41 ± 17.28 ^b^
*a* + *b* (g/kg DM)	933.01 ± 15.72 ^a^	409.45 ± 116.09 ^b^	804.56 ± 30.77 ^a^
ED (g/kg DM)	612.85 ± 23.25 ^a^	191.74 ± 10.65 ^c^	455.05 ± 42.93 ^b^
**NDF**			
*a* (g/kg DM)	87.82 ± 15.49 ^c^	162.08 ± 15.59 ^b^	214.19 ± 23.66 ^a^
*b* (g/kg DM)	788.51 ± 11.68 ^a^	414.21 ± 11.61 ^b^	458.06 ± 23.19 ^b^
*c* (g/kg DM)	87.76 ± 4.07 ^a^	20.56 ± 8.61 ^c^	54.43 ± 7.84 ^b^
*a* + *b* (g/kg DM)	876.33 ± 7.00 ^a^	576.29 ± 124.42 ^b^	672.25 ± 44.54 ^b^
ED (g/kg DM)	526.38 ± 3.03 ^a^	244.92 ± 10.83 ^c^	412.93 ± 18.68 ^b^
**ADF**			
*a* (g/kg DM)	134.26 ± 4.39	137.02 ± 22.41	102.44 ± 18.32
*b* (g/kg DM)	736.82 ± 35.47 ^a^	315.26 ± 49.50 ^c^	524.71 ± 12.98 ^b^
*c* (g/kg DM)	69.08 ± 5.28	58.41 ± 17.13	79.41 ± 2.79
*a* + *b* (g/kg DM)	871.09 ± 14.80 ^a^	452.27 ± 29.12 ^c^	627.15 ± 10.90 ^b^
ED (g/kg DM)	500.23 ± 14.13 ^a^	280.17 ± 20.50 ^c^	381.29 ± 14.32 ^b^

Values with different lowercase superscript letters (a, b, and c) are significantly different (*p* < 0.05) between different parts. ^1^
*a*: the rapid degradation fraction; *b*: the slow degradation fraction; *c*: the constant rate of degradation of *b* (%/h); ED: effective degradability; DM: dry matter; CP: crude protein; NDF: neutral detergent fiber; ADF: acid detergent fiber.

## Data Availability

The data presented in this study are available on request from the corresponding author. The data are not publicly available due to restrictions by the research group.

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
