# Peer review of "Effects of Different Parts on the Chemical Composition, Silage Fermentation Profile, In Vitro and In Situ Digestibility of Paper Mulberry"

_animals, 2021, doi:10.3390/ani11020413_

Round 1
Reviewer 1 Report
Dear Authors,
I have reviewed your manuscript 1033068 entitled “Effects of different parts on the chemical composition, silage fermentation profile, in vitro and in situ digestibility of paper” mulberry for the second time. Thank you for responding to my comments. I feel that the manuscript has considerably improved. There are a few point that I find important to consider.. These point are:
L55-61 This is irrelevant, please remove
L100 determination instead of detection
L176-181, Table 1, please add leaf : stem ratio
L176 comparing the whole plant weight with the components is irrelevant. Please remove
L187 Again you are comparing whole plant vs leaves and stems… so this is irrelevant… it is clear that whole plant = leaves + stems, so the comparison does not mean anything… Please remove from here and throughout the manuscript. It will be useful to describe whole plant silage quality and degradability without comparing it with leaves and stems…
L203 Table 3 does not display NH3 concentration. Also please indicate what is AN in table 3
L219-223 so this mean that leaves were degraded more than stems? If yes then please add
Through out the text, please add values of important variables (example DM%, ED of DM, NDF% ED NDF, CP and ED of CP) to the text to improve readability and give importance to these values
L265 have higher nutritional value rather than « more nutritional »
L282 remove (imperfection) and add (taken into consideration when PM-based diet is supplemented with concentrates)
L289-296 You have misread the study of Kung. At humidity 25%, pH of your legume silage should be 4.9 with elevated % of acetate (4.5% on DM basis) so it can be considered as well conserved. According to the data you presented in Table 3, the acetate level seem satisfactory in leaves silage, although the pH is elevated. There seem to be other unidentified chemical components in PM that conserve it under these conditions. Please use this explanation to clarify the findings, with the addition of findings of Si et al.
L332-337 After pointing out all the differences between leaves and stems in silage making and in in situ and in vitro degradability, it will be useful to conclude with recommendations like to mechanically separate leaves from stems before mixing with the diet, or recommending the use of varieties of PM that have higher leaves proportion in the plant….
Author Response
Dear reviewer
Thanks for your second comments. We have made revise based on your comments and highlighted them in green color in the manuscript. The revised detail could be seen in the attachment.
Thanks again for all the excellent advice to improve the quality of our paper.
Best wishes

Reviewer 2 Report
Authors have accepted the suggestions and the manuscript has been significantly improved since the first submission stage. I don’t have any further comment and I’m happy to inform that I’m positive about the publication of this manuscript in Animals, in case it’s also in accordance with Editorial Board.
Kind regards.
Author Response
Dear reviewer
Thanks for your positive comment on our paper. We have made a significant improvement after listening to your previous suggestions. We appreciated you again.
Best wishes
This manuscript is a resubmission of an earlier submission. The following is a list of the peer review reports and author responses from that submission.
Round 1
Reviewer 1 Report
Dear Authors,
I have reviewed the manuscript 875389 entitled Effects of different growth heights on the yield, chemical composition, silage fermentation profile, and in vitro digestibility of paper mulberry. Although I found merits in the study, I have concerns with statistical analysis, experimental methodology details and use of references. Please see my major and specific comments below.
Major comments
The manuscript is well written and easy to read through. The subject is important and of interest to scientific community and to local producers looking for locally produced forages. With such an elevated precipitation (600 mm/year), one would wonder what is the advantage of PM over other forages. Therefore a justification of interest in PM would add value to the manuscript.
Agronomical details need to be provided as soil type, fertilisation, temperature and precipitation during growth season, previous plant cover, seeding rate, land preparation details, equipement used for land preparation and seeding … please refer to a similar published article in any Agronomy Journal and provide similar details.
Experimental details on in vitro incubation are missing. What was the ratio of rumen fluid: buffer, what time was rumen fluid collected, what is the time between collection of rumen fluid and incubation? how was the rumen fluid treated and filtered after collection? 25 mL was collected from the three cows?....
A source of detailed methodology should be used when referencing techniques used. This should be done throughout the Materials and Methods section. For example, you refer to Zhang (23) for methodology of IVDMD measurement, and Zhang refers to another manuscript (Wang). This is not acceptable.
Statistical Analysis: There is no biological replication in the study. From the information provided in the Materials and methods, 20 plants were harvested for each GH, these 20 plants were pooled. Then the some plants (assume 7) were left as whole, while some plants (assume 13) were separated to leaves and stems. Then plant material were chopped to prepare silage bottles. So there was 1 pile of chopped whole plants, 1 pile of chopped leaves and 1 pile of chopped stems. True? There is no biological replication. If the 20 plants were from different fields, and you had 4 or 5 fields, then you would have biological replicates. It is not the case here. You had one sample that was divided in the lab, which is lab replicates. Therefore the statistical analysis was made on lab replicates and not biological replicates. This make the statistical analysis invalid for all the data.
Specific comment
L 20 did not increase to replace (had no increase)
L21 What do you mean by (fermentation quality degraded). Please replace with clearer expression
L33 same comment as above
L58 Discussing optimal harvest conditions of other plants is irrelevant. It is better to discuss what are the factors taken into consideration when harvest date-stage is chosen.
L69-70 This is unclear. Please reword
L73 Please add details on geographical location
L73 Is this the past 5 years average, if not then it should and it the time span of the average should be mentioned
L77 state interval between harvest in days
L84 remove “also”
L85 replace They with Samples
L85 what type of mill : hammer mill, rotating mill, ball mill please specify
L88 Van Soest
L89. Decrib the analysis of lignin separately was not analysed with ANKOM fiber analyser
L103 what is silage chemical detection? Is it composition? If yes then please modify
L104 why 24 h? is there a reference? If yes then please add it. Usually extractables are soluble in water immediately and can be adequetly measured after 30 minutes of mixing with water…
L108-109 Please use the original source of the analytical method
Table 1: Please add leaf: stem ratio
Table 1: Some values are not reasonable like lignin in whole plant GH 1.6: as GH increase, lignin is stable in leaves, but increasing in stems, then it should be increasing in W this is not the case. Also EE in W is higher or lower than EE in L or S which is impossible.
L155-157, As mentiond previously, this is mathematically impossible. Please check your data.
L157-159 here and in all manuscripts. Differences between leaves and stems in DM, CP and fiber is well established. Unless you have results different from the mainstream results, this discussion is deemed unnecessary. Please remove discussion of difference between leaves and stems throughout the manuscript.
Table 3 pH values of L and W are too high for a preserved silage (normal values should ≤ 4.5). This point to spoiled silage. Therefore measurements made on such silage samples would be invalid
Table 3 Lactic acid values for S are too high compared to WSC in table 1. How can a plant part with WSC at 8.59 g/kg DM produce a silage with lactic acid 91 g/kg DM. Please check your data.
L197-198 spell out parts A, B, C
L210 You are describing the chemical composition of 1 sample (20 plants) of PM to other crop. Which is not realistic. Please refer to my comment on statistical analysis.
L224-227 This is irrelevant because animals will not consume only the leaves, and will not consume PM in the diet (unless grazed). Ratio of Lys: Met in W is more relevant. Also more relevant discussion is comparing with other forages and possibility of incorporation in the diet.
L234-236. The concentration of NDF and ADF increase while WSC and CP decrease as GH increase, this provide less material for bacterial fermentation. Therefore the argument here is incorrect. Actually with higher DM content, less acid (from fermentation process) is needed to reduce pH to preservation level.
L266 comparison between leaf silage and silage of whole plants alfalfa or oat is irrelevant. A comparison between W and other forages would be relevant.
L278 good quality silage rather than qualified
Reviewer 2 Report
Suggestions and corrections were made based on PDF version with numbered pages and lines.
General points about the manuscript: The manuscript brings interesting information about different aspects of paper mulberry silage. In my opinion the manuscript has a good aim, it is well written, and authors used a good experimental design. I kindly pointed in this letter the issues that must be reviewed and fixed by the authors, or at least they should explain it better.
Specific considerations:
L28: Delete “was”.
Line 31: Delete “were” and “was”.
Line 32: contents, instead of content.
Line: 35 Replace “was” by “had”.
Keywords: Keywords will be used for indexing purposes and we should not use in keywords, those words already mentioned in the title. Use as keywords only words do not mentioned in the title.
L48: Please write “It was reported that the…”
L60: In my opinion, the sentences (There are various methods of conserving roughage, such as freezing and drying. These methods require more storage space, are costly and labour intensive) can be excluded.
L72: Do not start a sentence with an acronym (PM). And elsewhere along the manuscript.
L100: Was it really bottles? Or plastic bags instead?
L102: Is only 45 days enough for the proper and complete fermentation of the silage? Many authors are ensiling for at least 90 days.
Table 1: Instead of “groups”, please write “growth height”. For other Tables as well with the same layout.
L142: Growth height is not the only effect tested; parts of the plant is also another factor. So, please include this additional information to the title as well. And also for the other Table titles when it’s needed.
Tables: You have comparisons in a row (growth heights) and comparisons in a column (parts of the plant). You included SEM and P-value for the comparisons in a row, but these parameters were not included in comparisons in a column, why?
L150: Is it indeed “standard error of measurement”? Or is it “standard error of the mean”? These are different deviation measurements and it’s important to indicate the correct one. Using SAS, I would assume that the output provided the Standard Error of the Mean (and not standard error of measurement), please check and correct if needed. Same for other Tables.
L197: Please write “parameters” instead of “parameter”.
Table 4: The parameter A (theoretical minimum gas production) had P-value of 0.09, it’s above the significance level (P>0.05) indicated in M&M. So, if there is no statistical difference, why superscript letters were included? Or is the number 0.09 incorrect?
L229: …as well as…
L235: …which may be due…”
L238, 239: Do not start a sentence with an acronym.
L254: …its lower… instead of …its´ lower…
L263: The acetate and propionate acids are the main…
L274: With high protein…
L277: Do not start a sentence with an acronym. Do not use acronym in Conclusion.
L282: I would kindly suggest that at least S3 is normally included in the text to be published. These are very important data regarding chemical composition of the silage.
Best regards.
Reviewer 3 Report
General comments:
I evaluated the manuscript performed to assess the effect of growth height of paper mulberry on chemical composition, fermentative profile, and in vitro degradation of leaves, stem, and whole plant. Although the manuscript studies an interesting meaning, there are some relevant problems that the authors need to clarify. Firstly, the introduction lacks a clear hypothesis! Second, material and methods are terrible described. It is impossible to understand the treatments, experimental units, experimental design, and statistical analysis. The study seems to fail to use adequately experimental units. Third, the discussion needs to be rewritten.
Specific comments:
Lines 46-50: Why are you writing about mulberry fruit? Plants were harvested before the production of the fruits.
Lines 53-55: This study (8) evaluated PM silage and oat hay instead of corn silage and alfalfa hay in dairy cows diets. There was a substantial adverse effect on DMI, which was not mentioned in the present study.
Lines 57-60: Direct association with alfalfa and corn harvest is not correct because the material had growth behavior very different.
Lines 67-71: The authors need to state a clear hypothesis!
Material and methods
It is very confusing in this section. It is impossible to understand the treatments and experimental units were not defined. Also, statistical analysis re not adequately described.
Lines 73-80: What was the experimental unit? Was there repetition in the time or place that allows inferring that its results could repeat in other farms? Please, read https://doi.org/10.1016/j.anifeedsci.2011.10.008.
Line 101: What were the treatments? Why only three?
Lines 135-139: It is essential to provide a statistical model. Please, replace the means test by polynomial regression.
Line 142: Per plant?? Did you consider each plant as an independent experimental unit?
Line 151: Remove: “ The results of the yield and chemical composition of PM are presented in Table 1.”
Table 2. Unit is g/kg air-dried or g/kg of AA??
Table 3. How was this comparison among materials performed?
Line 191: Remove “The in vitro digestibility experiment results are shown in Table 4.”
The authors need to improve the discussion significantly. Some results, like those of AA profile, were not discussed: It was included only a brief review of most limiting AA.